# Increase of the particle hit rate in laser single particle mass spectrometer by pulse delayed extraction technology

Ying Chen[1,2], Viacheslav Kozlovskiy[3,5], Xubing Du[1,2], Jinnuo Lv[6], Sergei Nikiforov[4], Jiajun Yu[1,2], Alexander Kolosov[7], Wei Gao[1,2], Zhen Zhou[1,2], Zhengxu Huang[1,2], Lei Li[1,2]

[1] Institute of Mass Spectrometer and Atmospheric Environment, Jinan University, Guangzhou, 510632, China
[2] Guangdong Provincial Engineering Research Center for on-line source apportionment system of air pollution, Guangzhou, 510632, China
[3] Chernogolovka Branch of the N.N. Semenov Federal Research Center for Chemical Physics, Russian Academy of Sciences, Chernogolovka, 142432, Russia
[4] Prokhorov General Physics Institute, Russian Academy of Science, Moscow 119991, Russia
[5] Institute of Physiologically Active Compounds Russian Academy of Science, Chernogolovka, Moscow region, 142432 Russia
[6] Guangzhou Hexin Analytical Instrument Limited Company, Guangzhou, 510530, China
[7] N.N. Semenov Federal Research Center for Chemical Physics, Russian Academy of Sciences, Moscow, 119991 Russia

Correspondence to: Lei Li (lileishdx@163.com)

**Abstract.** Single particle mass spectrometer (SPMS) can provide a wealth of valuable information on chemical and physical parameters of individual particles in real time. One of the main performance criteria of the instrument is efficiency of particle detection (hit rate). Most of SPMS instruments use constant electrical field (DC) extraction, where stationary high voltage is applied to the extraction electrodes. As the aerosol particles initially carry a certain charge, those with a high amount to charge can be deflected by this electric field and lost, thus decreasing the hit rate. We realized that the delay extraction technique can eliminate the stochastic dispersion of the particle beam caused by their deflection in the stationary electric field. As the result, the hit rate of the instrument can be significantly improved. Also, as the effect of the deflection in the electric field is mass dependent, it can cause distortion of the measured size distribution of the particles. Hence, the delay extraction technique can bring the recorded distribution closer to the actual one. We found that the delay extraction technique provides mass resolution improvement as well as increases the hit rate. The gain in the hit rate depends on the type of particles. It can be two orders of magnitude for model particles, and up to 2-4 times for ambient particles. In the present work we report experiments and results showing the effect of the delay extraction on the beam divergence caused by particle charge, the hit rate improvement and the effect of the delay extraction on the measured particle size distribution.

## 1 Introduction

Aerosol particles have a strong impact on the climate, environment and human health. These effects are closely related to physical and chemical properties of the particles. An in-depth understanding of the physical and chemical properties of individual particles is important for studying various effects of aerosols. A Single Particle Mass Spectrometer (SPMS) is an

analytical tool that can provide the particle size and chemical composition of individual particles in real time. There are many publications that have reviewed the principle, structure and applications of SPMS in detail (Murphy, 2007; Nash et al.,

2006; Noble and Prather, 2000;Pratt and Prather, 2012). SPMS usually use an aerodynamic lens to focus and form an aerosol beam and to transfer the beam from the atmosphere into the vacuum system. As a common arrangement, a two-beam sizing laser measures the particle speed and size right before the "sized" particle passes into a high-power pulsed laser beam. When the particles are evaporated and ionized by the laser radiation, positive and negative ions are formed. These ions are accelerated by the strong electric field in the time-of-flight mass spectrometer (TOF MS) and then detected and recorded by

a data acquisition system based on a fast Analog-to Digital Converter card (ADC). Compared to the traditional off-line particle methods, the SPMS measurements can be conducted without preliminary collection and preparation of the particles providing rapid real time analysis. At present, SPMS is mainly used for characterization of physical and chemical properties of the particles, atmospheric chemical process analysis, atmospheric aerosol analysis and other environmental and material science applications(Bi et al., 2011; Kim et al., 2012; Li et al., 2014; Middlebrook et al., 2003; Roth et al., 2016). SPMS is

fast, and detection time for a single particle is typically on the order of a few milliseconds. However, it is common to accumulate single particle data over a period of time to obtain statistically significant information about particle diversity. The application of the SPMS to atmospheric aerosol analysis and investigation of chemical processes with atmospheric aerosols requires instruments with a high efficiency of particle detection (Hit rate). Hit rate is an important parameter of SPMS related to its performance. It is generally defined as the ratio of the number of spectra generated by laser ionization to

the number of particles detected by sizing lasers. There are many aspects that affect the hit rate of SPMS, including the particle size, physical and chemical properties (Dall'Osto et al., 2006; Zelenyuk et al., 2009) of the particle and the instrument design (Gemayel et al., 2016; Zhao et al., 2005).

It is well known (Dodd, 1953; Gunn and Woessner, 1956; Woessner and Gunn, 1956) that aerosol particles usually carry a certain amount of surface charge. If such particles enter the stationary electrical field in the extraction region of SPMS, they

can be deflected as they move through the ion source region. Once the particle trajectory has shifted from the center of the laser beam, the ionization efficiency can drop down noticeably (Su et al., 2004). Hence the particle deflection in the ion source results in a decrease of the hit rate. In order to reduce the influence of the electrical field on the particle beam trajectory, an electrostatic neutralizer can be installed between the particle introduction and particle ionization regions. The electrostatic neutralizer generally uses a radioactive source based on 210Po, 241Am, 85Kr, or an X-ray source (Demokritou

et al., 2004; Kousaka et al., 1981; Nicosia et al., 2018; Pratt et al., 2009; Tsai et al., 2005). However, the management of radioactive sources is a difficult task, the equipment is relatively expensive, and it is very inconvenient to use it in the field measurements because of strict regulations restricting their handling, transport, and storage (Nicosia et al., 2018).

Another problem in SPMS is particle beam divergence. There are many instruments that use Nd: YAG laser with 4[th] harmonic generation (266 nm wavelength) as the ionization source. Since the Nd:YAG laser with lamp pumping requires

more than 100 μs delay between the lamp ignition and the laser pulse, it is necessary to place the laser ionization source at a considerable distance (about 10 cm) downstream of the second particle sizing laser. The particle beam is dispersed on the

way from the sizing laser to the ionizing laser, and the increase of the beam width leads to a decrease in the hit rate. This problem can be solved using a laser with shorter delay time, such as excimer or nitrogen pulsed gas lasers (Alaime et al., 1983; Trimborn et al., 2002). It is well known that the delay time for the pulsed gas laser is less than 100 ns. The small delay

time makes it possible to reduce the distance between the sizing laser and the ion source, which will be limited by mass spectrometer design in this case. The effect can be bigger especially for irregular non-spherical particles (Zelenyuk et al., 2006). Shortening the distance between the sizing laser and the ionizing laser can compensate the influence of the particle shape to some extent (Cziczo et al., 2006; Thomson et al., 1997).

(Zelenyuk et al., 2009) also studied the shift of the trigger signals generated by particles with different sizes at the sizing

laser, which resulted in the hit rate being affected by the particle size. As a result, the dynamic laser trigger system was developed to implement the trigger compensation for the particles with different particle sizes thereby achieving an improvement of the hit rate. However, this method is directed to solve the positional spread of the aerosol particles along the axis; it does not contribute to the problem of lowering the hit rate caused by deflection of the aerosol in the direction perpendicular to the particle beam axis. This deflection is caused by static electrical field and depends on the ion source

dimensions and on the electrical field strength. Unfortunately, the ion source cannot be built short enough to neglect particle deflection. But, it is possible to switch off the electrical field by using a delay extraction technique (DE)( Li et al., 2018; Chudinov et al., 2019) for ion sampling.

In our previous works we have used the double exponential pulse delay extraction technique to improve the mass resolution of SPMS (Chudinov et al., 2019; Li et al., 2018). This solution provides mass resolution enhancement of about 2-3 times

over the broad mass range in comparison to the usual constant electrical field (DC) extraction. It is important that in the delay extraction ion source particles pass to the middle of the ion source with no DC electrical field. Therefore, the influence of the particle charge on its trajectory will be neglected in comparison with the usual ion source with a stationary electrical field. In this paper we present our investigation on the effect of stationary and pulse extraction on the particle trajectory and on the hit rate of the SPMS instrument.

**2 Instruments and methods**

The SPMS used in this study is a commercial instrument from Hexin Analytical Instrument Ltd. (Guangzhou), which is normally called a Single Particle Aerosol Mass Spectrometer (SPAMS). The detailed principle and design of the SPAMS has been described elsewhere in detail (Li et al., 2011). The commercial SPAMS was modified in two ways. First, the delay extraction technique is used instead of the original DC field extraction technology. The superposition of the rectangular and

the exponentially shaped extraction pulses simultaneously improves the resolution over the wide mass range for positive and negative ions. The detailed information on the instrument changes required to implement the delayed extraction is provided elsewhere (Li et al., 2018). After the improvement, a positive ions mass resolution of > 1000 FWHM and a negative ions mass resolution of > 2000 FWHM were achieved. The ssecond improvement is the use of multi-channel superimposed signal

acquisition technology. The native signal from the MCP detector is equally split in two channels with different amplification

ratios and acquired by two equal 8 bit ADC. Thus, the high dynamic range data acquisition system is capable of acquiring signals ranging from 5 to 20000 mV, and the dynamic range is nearly 40 times higher than that of the original SPAMS. This solution enables one, for instance, to detect high intensive alkali metal ions peaks together with a very weak ion signals from other elements or molecules at the same time(Shen et al., 2018).

The test particles used in the experiment were standard polystyrene latex microspheres (PSL) with particle sizes of 320 nm,

510, 720, 960, and 1400 nm purchased from Duke Scientific. The commercial aerosol generator TSI 9302 was used to atomize the aqueous solution of the PSL and to produce a controlled beam of particles. The air flow with PSL particles was dried by a diffusion drying tube and then passed to the SPAMS for analysis.

For an independent particle size distribution measurements the glass plate was installed under the ion source as shown in Fig.1. The distance between the centre of the ion source and the plate surface was 31.5 mm. After the glass plate was

exposed under the particle flow, the photo of the particles collected on the glass plate surface was acquired by an Olympus CX31RTSF microscope. The density profile of the collected particles was extracted by free *Image J* software. Trajectories of particles shown in Fig. 1 illustrate the deflection in the electrical field. The deflection depends on the charge and mass of the particle. The deflection and trajectory shape is discussed below in the Chapter 3.1

**2.1 Key factors affecting the efficiency**

Figure 1 shows a schematic representation of the SPAMS ion source. The Nd: YAG laser beam is focused in the center of the ion source where particles are ionized by laser radiation. The 266 nm laser operates in $TEM_{00}$ mode with Gaussian distribution. The $e^{-1}$ width of the focused Gaussian laser beam is about 300 μm. Factors that actually affect the ionization efficiency are the total laser pulse energy and the location of the particle in the focused laser spot. As the particle size is more than 300 times smaller than that of the focused laser beam width each particle is ionized by a virtually uniform laser power

density., But when the beam of the particles is wider than the diameter of the laser beam in the ionization region, the particles in the beam can be exposed to very different laser energy fluencies resulting in a great difference in ionization efficiency. Therefore, when the particles travel near the edge of the laser spot, it becomes impossible to generate enough ions to detect such particles.

In our experiments, the Ultra compact Q-switched Nd:YAG laser (Quantrel) was used with the pulse width 7.2 ns. The pulse

energy of the laser was set to 0.6 mJ, which corresponds to a laser power density in the focal point of about $2 \times 10^8$ Wcm$^{-2}$.

When the delay extraction technique is not used, the two DC high voltage potentials are constantly applied to the extraction electrodes of the original bi-polar SPAMS ion source. When the charged particles enter the space of the ion source, they start to be deflected under the action of the DC electrical field. The amount of deflection is related to the speed and the mass of the particles, the amount of the surface charge, and the electrical field strength. Thus, during the flight through the ion

source, the particles gradually deviate from the laser centre point, resulting in the decrease of the ionization efficiency and a significant drop in the hit rate.

When the delay extraction technique is used, the particles enter the extraction region while the electrical field is kept at zero. Then the laser ionizes the particle, and the HV extraction pulse is applied with 100 ns delay after the laser pulse. So there is no electric field between the ion source plates before the particles are ionized by the laser. Thus, the charged aerosol particles are not deflected by the electric field force during their flight, thereby achieving a higher hit rate.

## 3 Results and discussions

### 3.1 The experiment with PSL particles.

The influence of the electric field on the hit rate was studied by using PSL beads of five different particle sizes. In Fig.2 the dependencies of the hit rate from the electrical field strength for DC and DE mode are shown. It can be seen that under the same DC electric field strength (Fig.2 (A)), the hit rate of the particles from 320 nm to 1400 nm increases, indicating that the effect of the electric field on the small particles is more pronounced. As the DC electric field strength increases, the hit rate of the particles of each size gradually decreases, indicating that the particles are deflected by the electric field force.

Under the DE mode, the hit rate for the particles of all sizes is obviously increased in comparison with the DC case (Fig.2 (b)). When the pulsed electric field strength exceeds 60 kV m$^{-1}$, the hit rate of the particles above 520 nm is close to 100%, indicating that the flight path without an electrical field is essential for the increase of the hit rate of the particles. The hit rate of the 320 nm PSL particles is 40%, which is much lower than 100% of other particles. This can be due to the focusing effect of the aerodynamic lens. The divergence of the particle beam itself is higher than that for the particles above 500 nm, which results in a decrease of the hit rate. In addition, when the electric field strength increases from 0 to 60 kV m$^{-1}$, the hit rate gradually increases from 5% to 100%. We speculate that the increase of the hit rate with the electrical field strength in this range may be caused by the increase of the resolving power of TOF MS (Cotter, 1994). The lower is the extraction electrical field strength, the bigger is the turnaround time in TOF MS, which in turn decreases the resolving power of the instrument. In the case of a large turnaround time, the peak width increases and the peak height decreases. This, in turn, can result in a loss of measured ion current because of an ADC threshold adjusted to reject the low intensity peaks accounted as noise. The loss of the recorded ion current can in turn result in a miscount of some particles, thus decreasing the hit rate. Also, at the low extraction electric field, an ion divergence can be enhanced, resulting in a loss of ion current during the ion flight to the detector.

In the Fig.3 dependencies of the hit rate gain achieved for DE mode compared to DC mode case are shown. It can be seen that for different particle sizes, the hit rate gain increases sharply with the increase of the electric field strength, especially for small particle size. For 320 nm particles, the hit rate can be increased by more than 100 times at an electrical field strength over 180 kV m$^{-1}$. This result shows that the DE technique can effectively eliminate the influence of the electric field intensity on the flight path of the particles, avoid the use of additional peripheral devices such as electrostatic neutralizers, and result in an improvement of the hit rate.

It is worth noting that the mass spectra obtained with DE have a higher resolution, but it is approximately the same set of peaks as those obtained with the injection of ions in a constant field (CF). Peaks in the mass spectrum obtained with DE are narrower and, on average, have a larger amplitude. The influence of ~100 ns delay is not reflected in the composition of peaks in acquired mass spectra. Mass spectra of PSL particles acquired in DE and DC modes are presented in the supplementary material for comparison. The mechanisms of ion formation as part of a complex of processes during laser exposure to a particle are of great interest, and are the subject of our further research. In this work we did not focus on these issues, since they are not directly related to the observed increase of the hit rate.

In order to further confirm that the particles are deflected by the action of the electric field, we conducted several further experiments. Taking the 720 nm PSL microsphere as an example, we compared the hit rate of 720 nm PSL beads under the constant DC electrical field only, under the DC electric field with electrostatic neutralizer and delay extraction combined with or without a neutralizer. Fig.4 shows the hit rate value for a measurement period of ~30 minutes. It can be seen that the particle hit rate after the use of the electrostatic neutralizer has increased from an average level of 20% to a level exceeding 85%, indicating that the surface charge of the particles has a great influence on the hit rate. However, the average hit rate of the particles obtained by the DE technique is close to 96%, which is slightly higher than that of the hit rate measured in the case when the electrostatic neutralizer was used. This difference can be caused by a residual charge remaining on the particles even after the usage of an X-ray neutralizer (TSI 3088). The higher hit rate achieved by delay extraction is caused by the absence of electric field influence on the particle motion.

A further two experiments were performed to measure particle beam divergence under the influence of the electrical field formed between two electrodes of extraction region of SPMS. The particles were collected on the surface of the glass plate, as it is shown in Fig.1. In the first experiment (Fig.5 A,B) no voltage was applied to the electrodes, and the resulting spot just represents a 720 nm particle beam profile as a result of aerodynamic dynamic lens action. The profile was fitted by a Gauss envelope, and the average beam width extracted from the Gaussian curve is a little more than 0.1 mm. The second experiment (Fig.6 A, B) was performed when DC potentials (+/-1000 V) were applied to the extraction region electrodes according to the usual working conditions in the case of DC extraction mode. The trace of the same 720 nm PSL beads on the glass plate is shown in Fig.6 (A).

The particle divergence is obviously increased in this case. Using the ***Image J program***, the trace profile was extracted, and then the Gaussian fitting was uploaded on the data. The space distribution of the particles can be easily transformed into particle charge distribution as we deal with the calibrated PSL beads. The ion beam displacement at the exit of the extraction region (Fig.6 (A,B)) was analyzed by (Liu et al., 1995), and it can be described by the simple equation:

$$\Delta x_1 = \frac{1}{2}at^2 = \frac{1}{2}\frac{q}{m}\frac{V_H}{m_p d}\left(\frac{L_1}{U_p}\right)^2 \tag{1}$$

Here $q$ and $m_p$ are particle charge and mass, $d$ and $V_H$ are the distance and voltage difference between two flat extraction electrodes, $L_1$ is the electrodes height, $U_p$ is the particle longitudinal velocity, $a$ is the acceleration of the particle caused by the extraction electrical field and t is the particle dwell time in the extraction region. Thus, the particles trajectories are

parabolic, scaled by their mass and charge. The equation was derived by a simplified assumption that the particle moving between the plates is accelerated by the constant horizontal electrostatic force. Then the particle with the horizontal speed moves until it hits the gathering glass plate. The resulting particle displacement in the position of the gathering glass plate is:

$$\Delta x = \Delta x_1 + \Delta x_2 = \frac{1}{2}\frac{qV_H}{m_p d}\frac{L_1(L_1+2L_2)}{U_p^2} \quad (2)$$

Here $L_2$ is the distance from the lower edge of the extraction electrode to the gathering glass plate surface. The resulting particle displacement is proportional to the particle charge, so from the trace in Fig.6 (A) we can extract the particle charge distribution shown in Fig.6 (B). The average particle charge extracted from the Gaussian envelope is 60 elementary charges. There are multiple publications where this electrification charge was measured for different types of particles(Dodd, 1953). The results reported in the literature (40 charges per 700 nm road dust particle) (Forsyth et al., 1998), (40 charges per 400

nm PSL particle) (Kleefsman et al., 2008) are in reasonable agreement with our estimation made for 720 nm PSL beads.

Using known particle mass, velocity, extraction voltage and electrodes dimensions, we have extracted the particle displacement per unit charge in the middle point where the ionizing laser is focused on. For the parameters of our instrument this displacement is 12.5 $\mu m$ $z^{-1}$, where z is the number of elementary charges on the particle. Hence, a particle with an average of 20 elementary charges will be deflected ~250 μm from the ionizing laser spot centre. Taking into account the

Gaussian profile of the focused laser beam of 300 μm, the hit efficiency will be greatly affected by such displacement.

### 3.2 Comparison of environmental testing

The comparison of the detection efficiency for the instrument with DC extraction and with DE used for the actual atmospheric particulate matter detection is shown in Fig.7 (A,B).Aerosol particles from laboratory room air were analyzed by the SPMS used in this work. In order to ensure that aerosol composition is stable during the experiment, the DC and DE

modes of operation were switched alternately after 10 minutes.  Four sets of both DC and DE extraction data were acquired in 80 minutes. The hit rate was calculated as the ratio of the number of ionized particles to the number of sized particles. The hit rate data were extracted for the groups of particles selected by size; bin value was 20 nm while the whole range was 200-1000 nm. The data of four sets by 10 minutes each, both for DC and DE extraction modes, were summed. In the DC mode, in 40 minutes, a total of 12174 particles was recorded, of which 4569 were ionized, while in the DE mode, 12030 particles

were recorded, and 8228 were ionized for the same time. It is seen from Fig.7A, that the centre of the distribution obtained with DC extraction (Blue circles) is shifted to the higher particles size. This behaviour corresponds to our results obtained for the model PSL particles. Fig.7B shows the hit rate gain for the DE technique to the DC extraction, obtained from the data plotted in Fig.7A. It should be noted, that usage of delay extraction eliminates the distortion of particle distribution, which results from the dispersion of particle deflection by their size in case of DC extraction. The integral hit rate for the

experiment with the ambient particles is ~ 25% higher in case of delay extraction (Fig.7A), while the gain in hit rate achieved for small particles is ~ 4 times bigger than that measured for DC extraction (Fig.7B). The increased effect for small

particles can be explained by Eq. (1). The smaller the m/q of the particle, the bigger its deflection in the ion source (extraction region). Hence, the hit rate of smaller particles is dumped more, and the gain caused by DE should be higher.

The hit rate for the atmospheric aerosol is lower than that for the standard PSL beads mainly due to the shape of the particles.
In the actual atmosphere, the shape of the particles is much more complicated than the PSL sphere. For example, the aged carbon particles tend to be spherical, and the hit rate is relatively high, while the fresh black carbon particles exhibit a chain structure, so the divergence during their flight is probably bigger, resulting in lower hit rates(Ghan et al., 2012). In general, using the delay extraction technique, the hit rate of environmental particles with a size of < 500 nm can be noticeably improved, and the obtained particle size distribution becomes closer to the real one.

**4 Conclusion**

Aerosols often carry surface charges. When charged particles fly through the SPMS ion source, they can be deflected by the action of constant electric field, resulting in a substantial drop in the hit rate. When delay extraction is used in SPMS, the instrument efficiency is improved because the particles are not deflected by the action of the stationary electric field. The absence of the DC electrical field in this method allows one to avoid the use of the auxiliary external devices such as
electrostatic neutralizers, maximize the hit rate of the aerosol particles, and decrease the distortion of the measured size distribution of particles. Experiments have shown that for PSL microspheres, the smaller the particle size, the greater the gain in the hit rate. There is no dependence of the hit rate on the measured particle size for particles with a diameter of more than 500 nm when DE is used.  For the real application it means that this method minimizes losses in the hit rate, which in turn affects  the measured size distribution of the particles.

The experiments with environmental aerosol have also shown that SPAMS with delay extraction technology can improve the average detection efficiency of the actual atmospheric aerosol by more than 25%, and improve the hit rate for small (<500 nm) particles more than four times. It should be noted that the charge of the particles is connected with their parameters. It means that a pre-selection of particles based on their charge can be used to extract extra information from the data acquired by SPMS. It could be done, for example, by using two pairs of deflecting plates before the ion source. By choosing the
deflection voltage, only particles with a specific mass/charge ratio will pass this double deflector and can be ionized in the ion source.

**Author Contributions**

L.L. designed the study; Y.C., X.D. and L.L. performed the experiments; L.L.,V.K., S.N., X.D., Z.H. conducted the data analysis; L.L.,V.K., S.N. wrote the paper with the input from all authors

**Competing Interests**

The authors declare that they have no conflict of interest

**Acknowledgements.** This work was supported by the Guangdong Provincial Natural Science Foundation-Ph.D. Launched Vertical Collaborative Project(No.2017A030310384), Guangdong International Cooperation Project(No.2018A05056020), and National Key Research Project(No.2017YFC0209506). We would like to thank engineer Wang Jingjing of the Guangzhou Hexin Mass Spectrometer Co., Ltd. for his technical support, and Dr. Alexandre Loboda for the language proof.

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

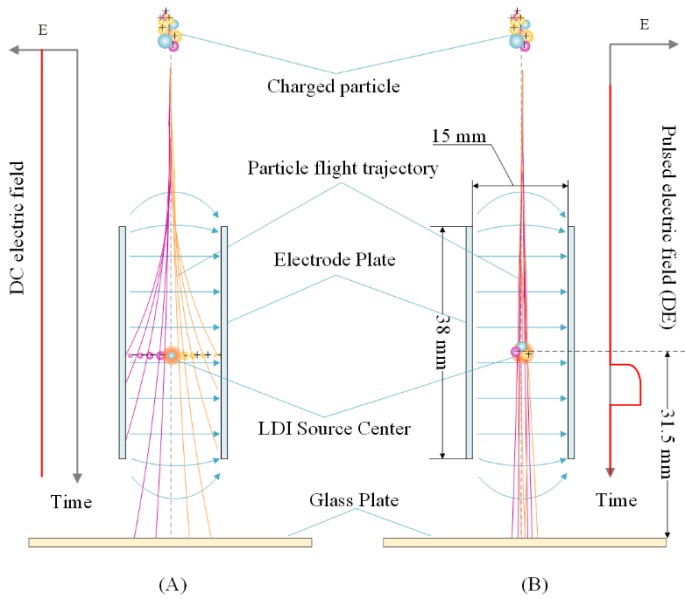

**Figure 1: Experimental setup. The glass plate position and extraction region dimensions are shown for two cases: (a) stationary electrical field extraction; (b) delay extraction. The size of the laser spot is ~ 300 μm**

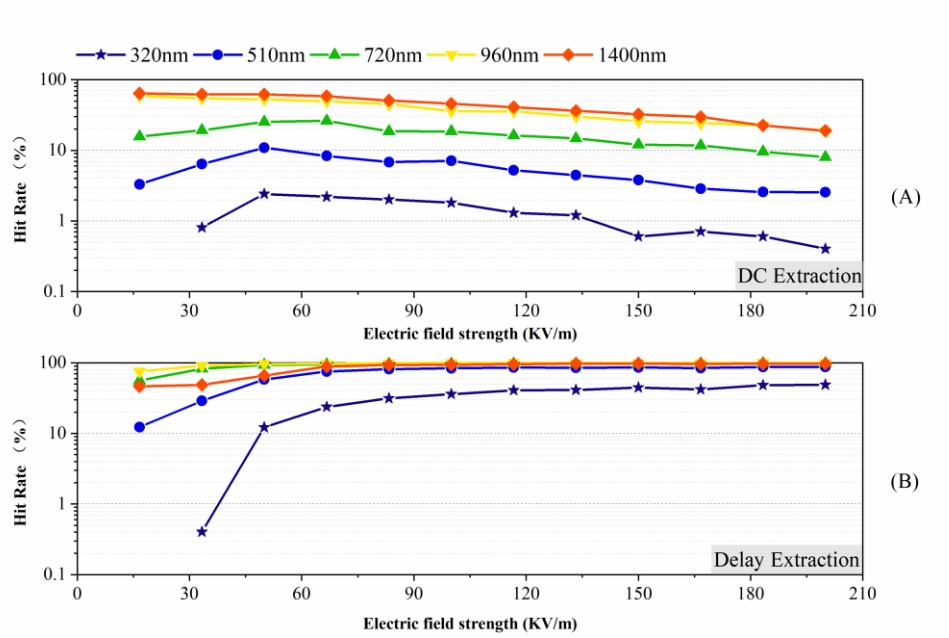

**Figure 2: Comparison of hit rate dependencies for different sizes of PSL particles in two cases: (A) constant electric field (DC extraction); (B) pulsed electric field (delay extraction, (DE)).**


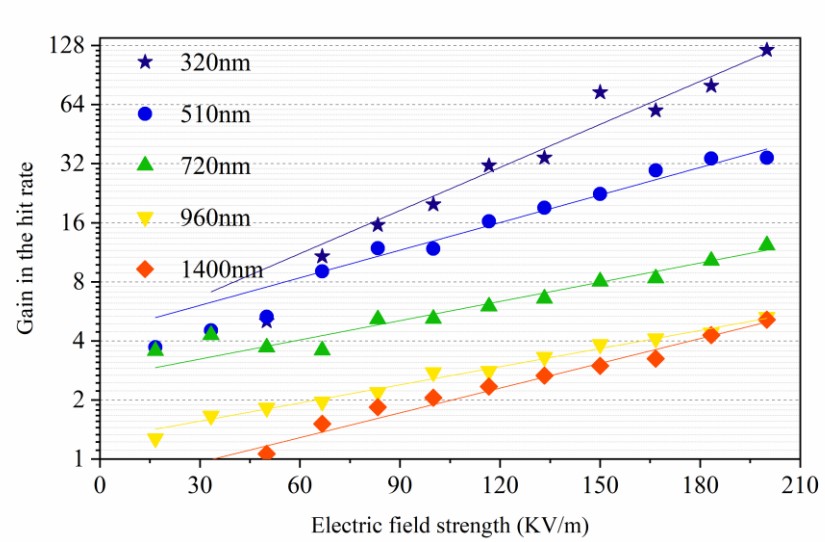

**Figure 3: The gain in the hit rate achieved for different sizes of PSL particles plotted over the extraction electric field strength.**

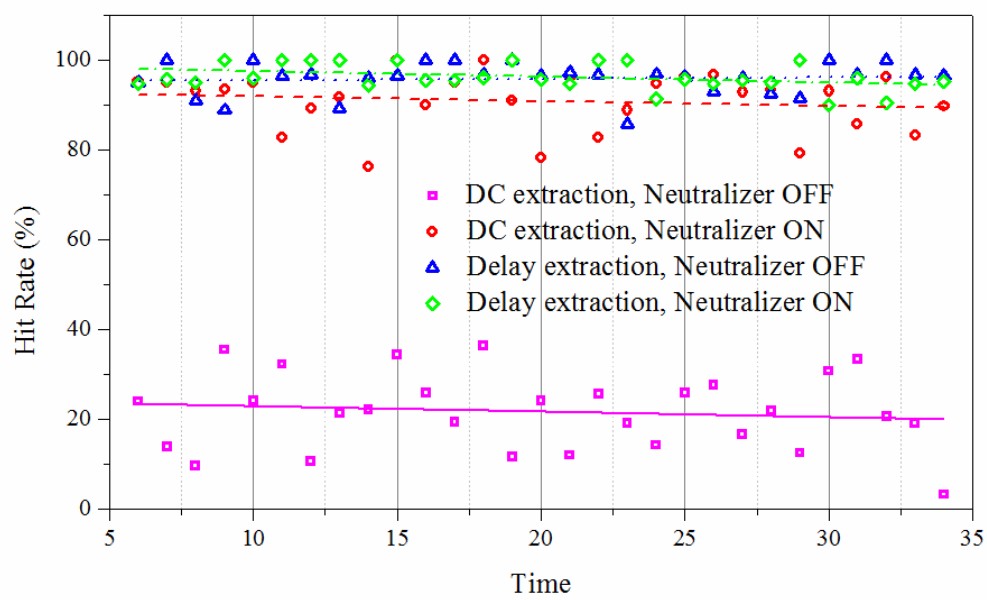


**Figure 4: Hit rate count for four cases: DC extraction& no neutralizer, DC extraction& neutralizer ON, Delay extraction & no neutralizer, and Delay extraction & neutralizer ON.**

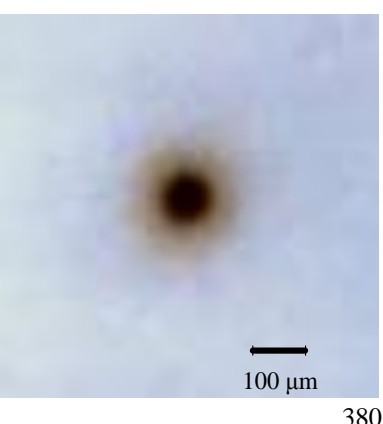

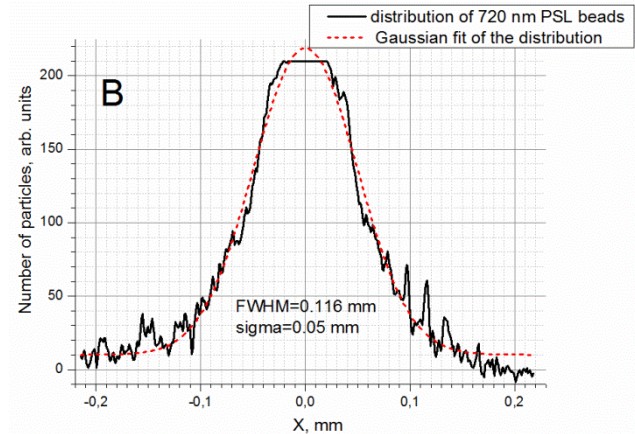

100 μm


**Figure 5: The distribution of 720 nm particles collected on the glass plate without electrical field. A – photo of the spot from particles collected on the glass plate. B – the density profile extracted from (A) by Image J software, and the Gaussian approximation of the profile.**


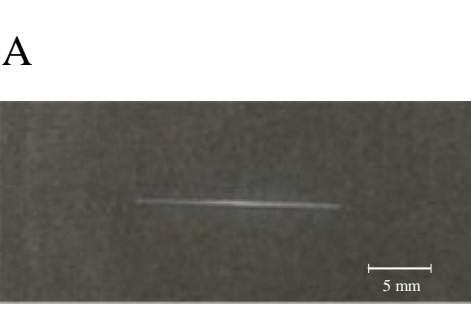

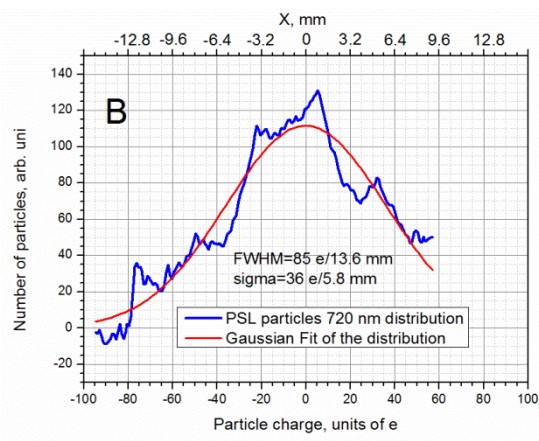

**Figure 6: The distribution of 720 nm particles moved through the electrical field formed with 2000 V potential difference between the electrodes in extraction region collected on the glass plate. (A) – photo of the trace from particles on the glass plate. (B) – the extracted particles charge distribution from the density profile (A) by Image J software, and the Gaussian approximation of the profile.**


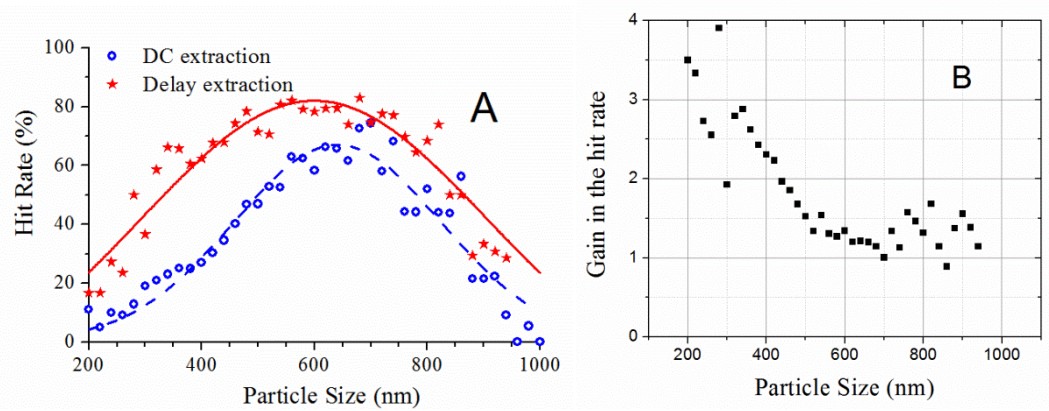

**Figure 7: A – Particles size distribution of ambient aerosol particles. Blue, circles - DC extraction, Red, asterisks - delay extraction; B – the hit rate gain achieved by delay extraction (DE) plotted over the particles size.**
