# Peer review of "Increase of the particle hit rate in laser single particle mass spectrometer by pulse delayed extraction technology"

_Atmospheric Measurement Techniques, 2019_

## Referee Comment (RC1) · Anonymous Referee #1 · 3 Jul 2019

This paper quantifies the improvement in SPMS hit rate resulting from particle beam deflection inside the electric field. It proposes that using delayed extraction helps to solve this problem and the results look convincing. I recommend the publication of this paper in AMT after the following major and minor comments are addressed.

Major comments: I would be really interested in seeing a comparison is mass spectra collected in DC and DE mode. In the DC mode, the ions produced by the laser are accelerated into the ToF immediately, but under the DE mode, they hang around the extraction region for an extra 100 ns, during which ion-ion and ion-neutral reactions can take place. It is possible that that would produce somewhat different fragmentation

patterns. I would appreciate if the authors gave some discussion of those possible effects.

Numerous English usage errors, especially with respect to article placement, inconsistent pluralization and subject-verb agreement. These usually don't inhibit understanding, but please have someone proofread it. I did not list them all.

Minor comments: p.2 lines 57-58: while high cost is a problem, I am not sure I agree that neutralizers are very inconvenient, especially compared to other aerosol instrumentation used routinely.

p.2 lines 59-60: I would also dispute that "most" instruments use Nd:YAG as the ionization source. There are quite a few prominent examples of the SPMS technology that use the excimer. In fact, there is a citation to Zelenyuk et al., (2006) paper in the same paragraph, which is describes an excimer-based instrument. It seems like your solution is primarily useful for instruments in which the distance between the sizing and ionization regions is large (i.e. ATOFMS or SPLAT), but not PALMS (sizing and ionization in the same physical instrument region). Differences in instrument design should be clarified here.

p. 3, line 69: you used "impact rate" instead of "hit rate" here, please be consistent.

Section 3.1 reads as though it belongs in the "Instruments and methods", since no actual data is presented yet.

Figure 4: why is there so much scatter in the DC measurements?

p.6: for completeness, it would help to show the same glass slide experiment in the DE mode. The divergence in the case of DC potential is dramatic. If the extraction pulse is applied instead, is the width of the beam at the slide significantly closer to the "voltages off" condition?

p. 7-8, lines 223-225: I am not sure I follow what experiment the authors have in mind by pre-selecting particles based on charge. I suggest this be fleshed out a bit more or

removed.

---

## Author Comment (AC1) · 5 Aug 2019

Dear Reviewer! Thanks a lot for your notes, comments and questions. We hope that they improve our understanding of the problems associated with processes in the single particle laser mass spectrometer. Please, find our answers/comments on your notes below:

1) Actually we don't observe a significant difference between mass spectral peaks in DE mode and DC mode, except their intensity and resolution. In our article we address other issues. i.e. the hit rate and the intensity of peaks in mass spectra. We have prepared the mass spectra acquired in DC and DE modes for comparison, for those

who are interested in this matter, and we will put them in the supplementary material. The plasma cloud after laser ionization is composed of ions, electrons and neutrals, and it is generally neutral. The plasma expands and disintegrates over time, and, at the same time, ion-neutral reactions and recombination processes between ions and electrons take place, so, the system is quite complicate. One can suppose, that if the effect of a strong electric field on the expansion of ions and electrons beyond 100 ns was significant, the signal in the case of DC extraction would be higher. On the contrary, we observe some increase in positive and negative ions signal in case of DE in comparison to the DC extraction mode. It can be assumed that the interaction of ions and neutrals makes a larger contribution than the recombination of ions with electrons. Presumably negative ions are formed by the capture of electrons by neutrals, while positive ions can be transformed into some cluster ions via interaction with neutrals. Thus, we can assume that the 100 ns delay can increase the interaction of charged particles with neutrals. It can also be assumed that the degree of ionization in the cloud is low. We upload three figures illustrating the comparison of mass spectra in DE and DC modes.

2) Thanks for the note about the language. We have corrected a number of errors by proof-reading.

3) We guess, the commercial X-ray neutralizer has two disadvantages, one is the price, it is more than 20,000 dollars if you want to buy a new one. The other is the X-ray lamp life time, which is only 8000 hours typically, and you need to change it frequently

4) About Nd:YAG – we agree, and we'll change the sentence in the article ("many instruments" instead of "most instruments")

5) Corrected to "hit rate"

6) We agree, it is better the section 3.1 should be transformed to 2.1. "Key factors affecting the efficiency" in the Instruments and Methods section.
7) The scatter can be related to the stochastic change in the surface charge of the particles. By the way, we have found a mistake in the Figure 4. Actually, it shows the hit rate per 1 second. In the manuscript, we described the whole time scale for this Figure as 30 minutes but actually it is 30 seconds. Because the hit rate was counted in every second, the particle detection rate is less than 10/second, so the result shows scattering according to the Poisson low. So we have change the minute into second in the Figure 4.

8) Unfortunately, it is hard to reproduce the experiment with the glass slide, as the instrument which was used for it is under construction now. But we don't suppose it will show any critical results, as the extraction pulse width is short (5 us), and the path of the particle moving with the velocity 100 m/s will be 0.5 mm. Hence, the effect of deflection will be  $\sim$  two orders of magnitude lower than in case of the DE. Hence, the spot size on the glass slide is supposed to be the same as that obtained without the electrical field (Fig.5), and we'll not get any extra information. Note please, that in case of DE operation, this HV pulse does not affect on the trajectory of the particular sized particle, as it is applied after it is ionized. On the other hand, the HV pulse is unlikely to affect other particles, since the average particle counting frequency is less than 100 Hz. Actually, the used 266 nm laser has a max repetition rate of 100 Hz, that is why the maximal counting frequency is 100 Hz.

9) We would add some more details about a possible realization to pre-select particles by their mass/charge ratio in front of the ion source: "It could be done, for example, by using two pairs of deflecting plates before the ion source. By choosing the deflection voltage, only particles with a specific mass/charge ratio will pass this double deflector and can be ionized in the ion source" It is just an idea which would need a significant additional work for it realization

Please also note the supplement to this comment: https://www.atmos-meas-tech-discuss.net/amt-2019-163/amt-2019-163-AC1-
**supplement.pdf**
**AMTD**

Fig. 1. Comparison of positive ions mass spectra obtained from PSL particles using SPAMS instrument with DC and DE extraction modes, in the mass range 0

**AMTD**

---

## Referee Comment (RC2) · Anonymous Referee #3 · 22 Sep 2019

Chen et al. descrbied a modified SPAMS to increase particle hit rate using a pulse delayed extraction technology. Indeed, the increase in hit rate is an improvement of instrumental performance. The attempt is exciting and novel. However, there are still concerns about the manuscript. Therefore, I recommend a major revision of the manuscript.

1. The delaying of ions, with an extra of 100 ns, can lead to secondary ion-ion and ion-neutral reactions. The artifact could cause inevitably shift of mass spectra, and the results could not be compared directed with current literature. Therefore, I would like to see some comparisons in both lab and field results between this new model and the

commercialized instruments. Related discussion is also necessary. 2. Increase of hit rate is undoubtedly an improvement of instrumental performance. However, single particle methodology is a partially sampling, meaning that the representation of full particle population is a major concern. The limited increase of hit rate and the unknown artifact, the balance should be considered cautiously. Again, the reviewer would like to see a discussion on this issue. 3. A serious proof-reading is necessary. 4. In Introduction, the reviewer recommends introducing a brief history of SPMS development. 5. The organization of the Method part should be improved. Please pay more attention to how your delaying system is designed. 6. Figure 2, it is not necessary to show the Y-axis in a logarithmic way, a linear one is enough. 7. Section 3.3, I would like to see some mass spectra of field (envrionment) particles. 8. Conclusion needs to be re-written. Please focus on what you did, the result, the benefit, and scientific implications. 9.

Miner revisions 1. Line 16, the full spell of DC is not provided. 2. "Delay extraction" is not proper because delay is used as a noun or a verb; "Delayed extraction" could be more appropriate. 3. Line 32, some high cited literature (Pratt and Prather, 2012) is not cited. 4. The space between paragraphs was not apparent; please add space into them.

Ref. Pratt, K.A., Prather, K.A., 2012. Mass spectrometry of atmospheric aerosols—Recent developments and applications. Part II: On-line mass spectrometry techniques. Mass Spectrom. Rev. 31, 17-48.

Please also note the supplement to this comment:
https://www.atmos-meas-tech-discuss.net/amt-2019-163/amt-2019-163-RC2-supplement.pdf

---

## Author Comment (AC2) · 19 Oct 2019

The comment was uploaded in the form of a supplement:
https://www.atmos-meas-tech-discuss.net/amt-2019-163/amt-2019-163-AC2-supplement.zip

---

## Author Response (AR1)

Dear Reviewers!

Thanks a lot for your notes, comments and questions. We hope that they improve our understanding of the problems associated with processes in the laser single particle mass spectrometer. Please, find our answers/comments on your notes below:

Anonymous Referee #1

This paper quantifies the improvement in SPMS hit rate resulting from particle beam deflection inside the electric field. It proposes that using delayed extraction helps to solve this problem and the results look convincing. I recommend the publication of this paper in AMT after the following major and minor comments are addressed.

Major comments: I would be really interested in seeing a comparison is mass spectra collected in DC and DE mode. In the DC mode, the ions produced by the laser are accelerated into the ToF immediately, but under the DE mode, they hang around the extraction region for an extra 100 ns, during which ion-ion and ion-neutral reactions can take place. It is possible that that would produce somewhat different fragmentation patterns. I would appreciate if the authors gave some discussion of those possible effects.

> Actually, we don't observe a significant difference between mass spectral peaks in DE mode and DC mode, except their intensity and resolution. In our article we address other issues. i.e. the hit rate and the intensity of peaks in mass spectra. We have prepared the mass spectra acquired in DC and DE modes for comparison, for those who are interested in this matter, and we will put them in the supplementary material.
>
> Actually, the plasma cloud after laser ionization is composed of ions, electrons and neutrals, and it is generally neutral. The plasma expands and disintegrates over time, and, at the same time, ion-neutral reactions and recombination processes between ions and electrons take place. If the effect of a strong electric field on the expansion of ions and electrons beyond 100 ns was significant, the signal in the case of DE would be weaker. On the contrary, we observe some increase in positive and negative ions signal in case of DE in comparison to the DC extraction mode. It can be assumed that the interaction of ions and neutrals makes a larger contribution than the recombination of ions with electrons. Presumably negative ions are formed by the capture of electrons by neutrals, while positive ions can be transformed into some cluster ions via interaction with neutrals.
>
> Thus, we can assume that the 100 ns delay can increase the interaction of charged particles with neutrals. It can also be assumed that the degree of ionization in the cloud is low. Below are three figures illustrating the comparison of

[Figure]

Fig. 1. Comparison of positive ions mass spectra obtained from PSL particles using SPAMS instrument with DC and DE extraction modes, in the mass range 0<m/z<120.

[Figure]

Fig. 2. Comparison of positive ions mass spectra obtained from PSL particles using SPAMS instrument with DC and DE extraction modes, in the mass range 120<m/z<220.

[Figure]

**Fig. 3. Comparison of negative ions mass spectra obtained from PSL particles using SPAMS instrument with DC and DE extraction modes, in the mass range 0<m/z<120.**

Numerous English usage errors, especially with respect to article placement, inconsistent pluralization and subject-verb agreement. These usually don't inhibit understanding, but please have someone proofread it. I did not list them all.

Thanks for the note about the language. We have corrected a number of errors by proof-reading.

Minor comments: p.2 lines 57-58: while high cost is a problem, I am not sure I agree that neutralizers are very inconvenient, especially compared to other aerosol instrumentation used routinely.

Actually, the commercial X-ray neutralizer has two disadvantages, one is the price, it is more than 20,000 dollars if you want to buy a new one. The other is the X-ray lamp life time, which is only 8000 hours typically, and you need to change it frequently

p.2 lines 59-60: I would also dispute that "most" instruments use Nd:YAG as the ionization source. There are quite a few prominent examples of the SPMS technology that use the excimer. In fact, there is a citation to Zelenyuk et al., (2006) paper in the same paragraph, which is describes an excimer-based instrument. It seems like your solution is primarily useful for instruments in which the distance between the sizing and ionization regions is large (i.e. ATOFMS or SPLAT), but not PALMS (sizing and ionization in the same physical instrument region). Differences in instrument design should be clarified here.

About Nd:YAG – we agree, and we'll change the sentence in the article (many instruments instead of Most instruments)

p. 3, line 69: you used "impact rate" instead of "hit rate" here, please be consistent.

Corrected to "hit rate"

Section 3.1 reads as though it belongs in the "Instruments and methods", since no actual data is presented yet.

We agree, the section 3.1 is transformed to 2.1. "Key factors affecting the efficiency" in the Instruments and Methods section.

Figure 4: why is there so much scatter in the DC measurements?

The scatter can be related to the stochastic change in the surface charge of the particles. By the way, we have found a mistake in the Figure 4. Actually, it shows the hit rate per 1 second. In the manuscript, we described the whole time scale for this Figure as 30 minutes but actually it is 30 seconds. Because the hit rate was counted in every second, the particle detection rate is less than 10/second, so the result shows scattering according to the $\sqrt{N}$ Poisson low. So we have change the minute into second in the Figure 4.

p.6: for completeness, it would help to show the same glass slide experiment in the DE mode. The divergence in the case of DC potential is dramatic. If the extraction pulse is applied instead, is the width of the beam at the slide significantly closer to the "voltages off" condition?

Unfortunately, it is hard to reproduce the experiment with the glass slide, as the instrument which was used for it is under construction now. But we don't suppose it will show any critical results, as the extraction pulse width is short (5 µS), and the path of the particle moving with the velocity 100 m/s will be 0.5 mm. Hence, the effect of deflection will be ~ two orders of magnitude lower than in case of the DE. Hence, the spot size on the glass slide is supposed to be the same as that obtained without the electrical field (Fig.5), and we'll not get any extra information.
    Note please, that in case of DE operation, this HV pulse does not affect on the trajectory of the particular sized particle, as it is applied after it is ionized. On the other hand, the HV pulse is unlikely to affect other particles, since the average particle counting frequency is less than 100 Hz. Actually, the 266nm laser has a max repetition of 100 Hz, that is why the maximal counting frequency is 100 Hz.

p. 7-8, lines 223-225: I am not sure I follow what experiment the authors have in mind by pre-selecting particles based on charge. I suggest this be fleshed out a bit more or removed.

We have added some more details about a possible realization to pre-select particles by their mass/charge ratio in front of the ion source. It is just an idea which would need a significant additional work for it realization

**Anonymous Referee #3**

Chen et al. described a modified SPAMS to increase particle hit rate using a pulse delayed extraction technology. Indeed, the increase in hit rate is an improvement of instrumental performance. The attempt is exciting and novel. However, there are still concerns about the manuscript. Therefore, I recommend a major revision of the manuscript.

1. The delaying of ions, with an extra of 100 ns, can lead to secondary ion-ion and ion-neutral reactions. The artifact could cause inevitably shift of mass spectra, and the results could not be compared directed with current literature. Therefore, I would like to see some comparisons in both lab and field results between this new model and the commercialized instruments. Related discussion is also necessary.

The objective of this work is to demonstrate an increase in hit rate when applying Delayed Extraction (DE). In our opinion, this is demonstrated very convincingly, and a physical explanation of the observed effect is given in the work. This effect depends only on the particle charge and the presence of a static electrical field. Particle charge is an absolutely common property for both laboratory and natural particles. The ion formation processes during laser desorption/ionization of a particle is not considered in detail in this paper, and therefore, comparison with other devices by the type of spectra obtained is hardly appropriate here. By the way, we presented such a comparison of mass spectra of PSL particles in the supplement. As we noted in the first answer, there is a noticeable effect of DE on the shape of the spectrum, which is due to the features of the operation of time-of-flight mass spectrometers, and, possibly, reactions in a cloud of desorbed molecules. The mechanisms of ion formation as part of a complex of processes during laser exposure to a particle are of great interest, and are the subject of our further research. But in this publication we did not go deep into these issues, since they are not directly related to the observed effect, and their detailed exposition requires a much larger volume.
     There are two other considerations:
1) we have the initial ion velocity spread after laser desorption / ionization of a particle. Due to the delay, we transform the velocity spread into the coordinate ion spread (time-lag focusing), which is correlated with the velocity spread. If ions significantly undergo ion-molecular reactions within a 100 ns delay, the resulting coordinate spread after the delay will be not correlated with the velocity spread any more. And we would not receive a gain in resolution. But we get it.
2) the particle composition after the laser shot is electrically neutral. That is, recombination reactions between positive and negative ions can occur. The rate constant of this process is in any case no less than that of the reactions between ions and neutrals. Then, if such reactions had a significant effect, then we would see a decrease in the signal in the case of DE due to recombination/neutralization of a part of the ions. We do not see it, but rather, on the contrary, we observe a slight increase in the signal; hence, by our mind, the reactions of the formed ions do not particularly change in the case of DE.

 2. Increase of hit rate is undoubtedly an improvement of instrumental performance. However, single particle methodology is a partially sampling, meaning that the representation of full particle population is a major concern. The limited increase of hitrate and the unknown artifact, the balance should be considered cautiously. Again, the reviewer would like to see a discussion on this issue.

Indeed, particles observed by single-particle MS are a partial statistical sampling from an aerosol array. This sampling is carried out in several steps during analysis in the MS. These are the transmission of the aerodynamic system, the hit rate in the ion source, the work of which is devoted to the increase, and the processes of formation and registration of ions. In our opinion, an increase in hit rate brings the sampling closer to the total aerosol array in terms of the number and size distribution of the detected particles

3. A serious proof-reading is necessary.

OK, we'll do it once again, and we'll correct the manuscript

4. In Introduction, the reviewer recommends introducing a brief history of SPMS development.

We added the reference [Pratt, K.A., Prather, K.A., 2012. Mass spectrometry of atmospheric aerosols. Recent developments and applications. Part II: On-line mass spectrometry techniques. Mass Spectrom. Rev. 31, 17-48] in the introduction section.

5. The organization of the Method part should be improved. Please pay more attention to how your delaying system is designed.

We have reported the detail of the time-delay system  on the article "Improvement in the Mass Resolution of Single Particle Mass Spectrometry Using Delayed Ion Extraction". (https://link.springer.com/article/10.1007/s13361-018-2037-4)

6. Figure 2, it is not necessary to show the Y-axis in a logarithmic way, a linear one is enough.

In our opinion, a plot with Y range of 3 orders of magnitude is hardly shown in a linear way.

7. Section 3.3, I would like to see some mass spectra of field (envrionment) particles.

We presented some mass spectra in the supplementary material for comparison. . We plan to make a detailed comparison in future work in relation with the study of ionization mechanisms.

8. Conclusion needs to be re-written. Please focus on what you did, the result, the benefit, and scientific implications.

We'll try to revise the conclusion.

 9. Miner revisions 1. Line 16, the full spell of DC is not provided.

OK, we updated the sentence

 2. "Delay extraction" is not proper because delay is used as a noun or a verb; "Delayed extraction" could be more appropriate.

Thanks, we corrected the sentence

3. Line 32, some high cited literature (Pratt and Prather, 2012) is not cited.

The reference added in the Introduction section

4. The space between paragraphs was not apparent; please add space into them.

The formatting is updated

Ref. Pratt, K.A., Prather, K.A., 2012. Mass spectrometry of atmospheric aerosolsâ˘ AˇTRecent developments and applications. Part II: On-line mass spectrometry techniques. Mass Spectrom. Rev. 31, 17-48.

Please also note the supplement to this comment: https://www.atmos-meas-tech-discuss.net/amt-2019-163/amt-2019-163-RC2supplement.pdf

[revised manuscript text omitted]

---

## Author Response (AR2)

Dear Editor, dear Reviewer!

Thanks a lot for the useful comments; we hope they will improve the manuscript. In most cases the corrections are made relating to the notes, and some citations are updated. The Supplementary material is now prepared as the separate file. We also have changed some authors affiliations because during the drawn out revision process some Institutions were re-organized **(Lines 4,8,9,15)**. Also, the models of the laser and neutralizer are added in Lines 125,181.
Please, find below our answers (in violet) made through the reviewer text.

Best Regards,
Authors of the manuscript Amt-2019-163

General Comments:
This paper describes the addition of delayed extraction to the ion source of a single particle mass spectrometer. The authors investigate the hit rate of particles, i.e. the percent of particles that are sized by the mass spectrometer system from which viable mass spectra are generated. The authors make the case that the implementation of delayed extraction is simpler than the use of an external particle neutralizer (see comments below) and that the hit rate improvement is significant. Overall, the paper shows relatively dramatic impacts on hit rates of polystyrene latex spheres (PSLs) and modest impacts on indoor particles of mixed composition. The authors provide no information in this paper about the instrument changes that are required to implement delayed extraction, including addition of power supplies, adjustments to instrument timing circuitry, etc., and, while they make a good case for the well-known impact of particle charge on particle trajectories in electric fields, they do not make their case that delayed extraction is simpler than the use of a neutralizer.

We wouldn't like to blow up the article size with the detailed instrumentation description presented elsewhere recently. The detailed information on the instrument changes required to implement delayed extraction is provided in the article referenced in the manuscript (Li et al., 2018, Li, L., Liu, L., Xu, L., Li, M., Li, X., Gao, W., Huang, Z., and Cheng, P.: Improvement in the Mass Resolution of Single Particle Mass Spectrometry Using Delayed Ion Extraction, Journal of the American Society for Mass Spectrometry, 29, 2105-2109, http://doi.org/10.1007/s13361-018-2037-4, 2018). One sentence is added in Line 97-98

In the article, we would like to demonstrate that the use of DE allows us to not only increase the resolution and dynamic range of the instrument (Li et al., 2018), but also noticeably increase the hit rate. Sure, the physics of the process is clear. But the magnitude of the effect was not investigated experimentally or theoretically, so the results obtained with the model and the ambient particles seem to be useful. The discussion of the complexity of using a DE or a neutralizer depends on the particular instrument design and, in our opinion, is hardly appropriate in the article.

Specific Comments:

1. The English language usage in the manuscript is not yet grammatically clean enough for publication. There are too many issues to enumerate here, but revision by a native speaker is imperative.

We have revised the manuscript with the native speaker. Mostly, the articles are corrected in multiple places,
Lines17,19,20,22,27,29,33,37,60,64,79,85,88,90,93,98,99,100,101,104,109,111,112,124,126,130,141,146,148,151,156,157,168,170,174,175,176,177,181,182,183,185,186,187,189,191,193,194,199,211,212,213,219,221,239,247,248

2. The authors refer to figures in the supplementary information (line 161), but no supplementary information was available with this manuscript to review.

The Supplementary Information file was occasionally appeared in the cumulative answer to the discussion reviews. Now it is uploaded separately as Supplementary file.

3. The authors state (lines 58 – 60) that "the management of radioactive sources is a difficult task, the equipment is relatively expensive, and it is very inconvenient to use it in the [sic] field measurements." However, they give no explanation of the costs, hardships, or complexity, nor do they provide this information in comparison to their proposed solution. It is not sufficient to state that something is better in specific ways without providing information about both methods. It is likely that the challenges associated with radioactive neutralizers are location (i.e. regulation) specific. The use of X-ray neutralizers, however, is typically more straightforward. Cost comparisons of the neutralizers with the modifications made to the commercial instrument (additional power supplies, software modifications, etc.) are not discussed.

We talk about neutralizers in the Introduction section describing available techniques for single particle measurements, and just claim their features, also reported by other researchers. The particular comparison of the costs, hardship, complexity of our solution with the neutralizers was not an aim of our work. We believe it is hardly worth to include it into the article. One sentence is added in the Line 63.

4. The majority of the work done in this paper is using spherical PSL particles, with a small experiment added at the end in which indoor aerosols are sampled. The impact of the delayed extraction on the PSLs is shown to be 1 – 2 orders of magnitude, depending on field strength (shown in Figure 2), whereas it is only 2 - 5x greater for room-air particles below 500 nm and only 0.25x greater for room-air particles ≥ 500 nm in diameter. The authors state that this difference can depend on composition, but make no effort to explain their results using composition information, which they have from the SPMS instrument. This portion of the paper should be significantly enhanced to make a stronger case for the utility of this instrumental modification for the analysis of ambient aerosol particles.

The difference in the impact of delayed extraction can be explained by different surface charge carried by particles having different composition and different shape. As ambient particles is the complex

mixture of different particles, the detailed consideration of the effect looks too speculative. Also, we suppose, that m/q is lower for the smaller particles, resulting stronger deflection in the electrical field. The short sentence is added in Line 231-234

A side-note, the use of the term "real" aerosol particles (line 221) is incorrect. All particles sampled are real, but only the ones sampled from the room are "ambient" or "not lab generated."
Thanks for the note. The «real» is substituted by «ambient» in the Line 27, 230, 399.

5. Throughout the paper, the authors state but do not illustrate that the issue they are trying to solve is "significant." It is not until the very end of section 3.2 that quantitative information is provided about particle deflection relative to the size of the laser spot. This is the fundamental issue that they are trying to solve and quantitative information about it is not provided to make the case at the beginning. In addition, in line 124, the authors state that this dispersion leads to a "significant" drop in hit rate, without giving any definition of the use of the word. I would recommend that they not use the term "significant" since it carries with it specific criteria that they are not addressing.
Thanks for the note.
The words «significant» are substituted by another ones in Lines 57, 191, 240,

6. Figure 1 shows a schematic of particle deflection but it is not clear whether the dispersion illustrated for the particle beams as drawn is based on any calculations, or whether it is just a cartoon to show the possible impact of the electric field on the particle beam. This must be clarified and the figure should show quantitative information if at all possible. Including the laser spot size would also be helpful.

The comments are added in line 113-114. The details of the laser spot size are reported in Line 118. The shape of the particle trajectory is reported in Line 198-199. The consideration of the particle deviation is available in Lines 231-234

7. The authors mention early on that the beam divergence problem "can be solved" using lasers with shorter delay times, but they give no information about this.

The explanation is added in Lines 70-72.

8. In the "Instruments and methods" section, lines 94 – 98, the authors describe a modification made to their instrument that replicates that in use in laboratory and commercial ATOFMS instruments for many years. Citations are necessary.

They were presented, (Li et al., 2018) and (Shen et al., 2018), Lines 98, 104

9. In lines 141 – 143, the authors state that the poor performance for 320 nm PSL particles is due to the aerodynamic lens' characteristics. However, they provide no data about the performance of the lens to support this.

This statement is our assumption. Unfortunately, we have no possibility to measure the performance of the particular aerodynamic lens now. Therefore, the phrase is replaced by "can be due…" ( Line 149)

10. Given the size range that this instrument is stated to cover, and the distribution shown for the room-air particles in Figure 7, it is surprising that the authors didn't attempt to quantify the impact of delayed extraction on particles between 200 and 320 nm. This is likely because atomization of particles below approximately 300 nm tends to produce artefacts, and therefore they are typically generated using methods that incorporate neutralizers in the system (such as a DMA). However, this should be discussed, as the results presented suggest that the issue addressed by incorporation of delayed extraction is more important for smaller sized particles.

We are studying this issue, but in the presented work we just report the clear physical effect and confirm it by a few experiments. The short consideration is added in Lines 231-234.

Technical Details:
1. Terms are not defined or are used inconsistently:
a. ADC (line 38 and below) The definition is added in Line 41
b. Turnaround time (line 146 and below) Reference added in Line 154
c. Hit rate increment (line 152 and below) Changed to "Hit rate gain", Line 160
d. Constant field (CF) (line 159) is used when DC is the abbreviation defined in the abstract. It should be defined in the body of the text, as well. The definition of DC is added in Line 86
2. What the authors refer to as "light laser calipers" (line 35 and below) is typically called aerodynamic particle sizing in the aerosol literature. Revised in Lines 38, 51, 68, 71, 73, 76
3. The authors state that the types of experiments for which SPMS instruments are used typically "requires a high performance instrument" (line 45) but give no indication what they mean by this. Revised to "high efficiency of particle detection", Line 49
4. The authors claim that their results are "in good agreement" for the charge on the 720nm PSLs, but they do not provide any quantitative information about what is predicted in the source they site. Their value should be compared. Two references and reference values are added in Lines 207-208

The list of the changes in the manuscript
1. Changed affiliations because during the drawn out revision process some Institutions were re-organized (Lines 4,8,9,15).
2. The models of the laser and neutralizer are added in Lines 125,181.
3. One sentence is added in Line 97-98 about details of DE implementation
4. One sentence is added in the Line 63 about neutralizers
5. The short sentence is added in Line 231-234 about the DE effect on different particles
6. The «real» is substituted by «ambient» in the Line 27, 230, 399.
7. The words «significant» are substituted by another ones in Lines 57, 191, 240

8. The comments are added in line 113-114. The details of the laser spot size are reported in Line 118. The shape of the particle trajectory is reported in Line 198-199. The consideration of the particle deviation is available in Lines 231-234

9. Different lasers synchronization short sentence is added in Lines 70-72

10. The phrase is replaced by "can be due…" ( Line 149)

11. The ADC definition is added in Line 41

12. Turnaround time reference is added in Line 154

13. «Hit rate increment» is changed to "Hit rate gain", Line 160

14. The definition of DC is added in Line 86

15. «Laser caliper» is changed to «sizing laser» in Lines 38, 51, 68, 71, 73, 76

16. «High performance Instrument» is revised to "high efficiency of particle detection", Line 49

17. Two references and reference values are added in Lines 207-208

18. Multiple articles, commas are changed according to the English proof in Lines17,19,20,22,27,29,33,37,60,64,79,85,88,90,93,98,99,100,101,104,109,111,112,124,126,130,141,146,148,151,156,157,168,170,174,175,176,177,181,182,183,185,186,187,189,191,193,194,199,211,212,213,219,221,239,247,248

[revised manuscript text omitted]